# Network Pharmacological Analysis through a Bioinformatics Approach of Novel NSC765600 and NSC765691 Compounds as Potential Inhibitors of *CCND1*/*CDK4*/*PLK1*/*CD44* in Cancer Types

**DOI:** 10.3390/cancers13112523

**Published:** 2021-05-21

**Authors:** Ntlotlang Mokgautsi, Yu-Chi Wang, Bashir Lawal, Harshita Khedkar, Maryam Rachmawati Sumitra, Alexander T. H. Wu, Hsu-Shan Huang

**Affiliations:** 1PhD Program for Cancer Molecular Biology and Drug Discovery, College of Medical Science and Technology, Taipei Medical University and Academia Sinica, Taipei 11031, Taiwan; d621108006@tmu.edu.tw (N.M.); d621108004@tmu.edu.tw (B.L.); d621108005@tmu.edu.tw (H.K.); d621109006@tmu.edu.tw (M.R.S.); 2Graduate Institute for Cancer Biology and Drug Discovery, College of Medical Science and Technology, Taipei Medical University, Taipei 11031, Taiwan; 3Department of Obstetrics and Gynecology, Tri-Service General Hospital, National Defense Medical Center, Taipei 11490, Taiwan; yuchitsgh@mail.ndmctsgh.edu.tw; 4TMU Research Center of Cancer Translational Medicine, Taipei Medical University, Taipei 11031, Taiwan; 5The PhD Program of Translational Medicine, College of Science and Technology, Taipei Medical University, Taipei 11031, Taiwan; 6Clinical Research Center, Taipei Medical University Hospital, Taipei Medical University, Taipei 11031, Taiwan; 7Graduate Institute of Medical Sciences, National Defense Medical Center, Taipei 11490, Taiwan; 8School of Pharmacy, National Defense Medical Center, Taipei 11490, Taiwan; 9PhD Program in Drug Discovery and Development Industry, College of Pharmacy, Taipei Medical University, Taipei 11031, Taiwan

**Keywords:** drug resistance, cancer stem cells (CSCs), drug-likeness, pharmacokinetics, bioinformatics

## Abstract

**Simple Summary:**

Around 14 million new cancer cases, rate are reported annually, with high mortality worldswide, several mechanisms are associated with complexities in cancer, which leads to resistance to current therapeutic interventions in cancer patients. The aim of this study was to identify molecular genes responsible for cancer development, progression and resistances to therapeutic intervention, and also evaluate the potency of our novel compounds NSC7565600 and NSC765691 as potential target for these oncogenes. Using bioinformatics, we successfully identified *CCND1/CDK4/PLK1/CD44* as oncogenic signatures, which drives cancer progression and resistance to treatment, and as potential druggable candidates for both NSC7565600 and NSC765691 small molecules. We also showed the antiproliferative and cytotoxic effects of these compounds against a panel of NCI-60 cancer cell lines. This suggests the potential of NSC765600 and NSC765691 compounds to inhibit *CCND1/CDK4/PLK1/CD44* expressions in cancer.

**Abstract:**

Cyclin D1 (*CCND1*) and cyclin-dependent kinase 4 (*CDK4*) both play significant roles in regulating cell cycle progression, while polo-like kinase 1 (*PLK1*) regulates cell differentiation and tumor progression, and activates cancer stem cells (CSCs), with the cluster of differentiation 44 (*CD44*) surface marker mostly being expressed. These oncogenes have emerged as promoters of metastasis in a variety of cancer types. In this study, we employed comprehensive computational and bioinformatics analyses to predict drug targets of our novel small molecules, NSC765600 and NSC765691, respectively derived from diflunisal and fostamatinib. The target prediction tools identified *CCND1/CDK4/PLK1/CD44* as target genes for NSC765600 and NSC765691 compounds. Additionally, the results of our in silico molecular docking analysis showed unique ligand–protein interactions with putative binding affinities of NSC765600 and NSC765691 with *CCND1/CDK4/PLK1/CD44* oncogenic signaling pathways. Moreover, we used drug-likeness precepts as our guidelines for drug design and development, and found that both compounds passed the drug-likeness criteria of molecular weight, polarity, solubility, saturation, flexibility, and lipophilicity, and also exhibited acceptable pharmacokinetic properties. Furthermore, we used development therapeutics program (DTP) algorithms and identified similar fingerprints and mechanisms of NSC765600 and NSC765691 with synthetic compounds and standard anticancer agents in the NCI database. We found that NSC765600 and NSC765691 displayed antiproliferative and cytotoxic effects against a panel of NCI-60 cancer cell lines. Based on these finding, NSC765600 and NSC765691 exhibited satisfactory levels of safety with regard to toxicity, and met all of the required criteria for drug-likeness precepts. Currently, further in vitro and in vivo investigations in tumor-bearing mice are in progress to study the potential treatment efficacies of the novel NSC765600 and NSC765691 small molecules.

## 1. Introduction

Cancer is one of the most prevalent and deadliest diseases globally, with an incident increase of approximately 14 million new cancer cases annually [1], and the total number of patients anticipated to be 450 million by 2025 [2]. There are several mechanisms associated with complexities in cancer, including survival strategies implemented by cancer cells to escape from cytotoxic therapies, which consequently lead to resistance to current therapeutic interventions [3]. In addition, immunotherapy has evolved as a new approach in oncology, and patients exhibit greater tolerance to this approach than to traditional alternative medicines [4]. However, cancers in most patients have been reported to be resistant to immuno-checkpoint inhibitors (ICIs) after a certain time [5]. Accumulating evidence has shown that another mechanism through which cancer develops and progresses results from changes in the cell cycle [6]. Recent studies have demonstrated that cyclin D1 (*CCND1*) plays a significant role in regulating the cell cycle, thereby promoting tumor proliferation, invasion, and metastasis [7], as well as angiogenesis and resistance to chemotherapy and radiotherapy in multiple cancer types [8,9]. Moreover, *CCND1* overexpression has been reported in several cancers including lung cancer, breast cancer, colon cancer, glioblastomas, melanomas, and oral squamous cell carcinoma, with an amplification rate of approximately 40% [10]. This is associated with metastasis, negative responses to ICIs, and poor prognoses of solid tumors [11,12,13].

As an oncogene, *CCND1* was shown to promote tumor growth by regulating cyclin-dependent kinase 4 (*CDK4*), through modulating the cell cycle transtion from the G_1_ to the S phase, thereby making *CCND1* and its regulatory partner, *CDK4,* attractive potential therapeutic targets [14]. *CCND1* treatment includes drugs that mainly target its transcription and protein synthesis; however, these current treatment options are still limited due to the resistance that eventually develops [15]. Therefore, novel therapeutic approaches are needed that give rise to better responses. Since *CDK4* is activated by *CCND1*, recent studies have shown that it is also overexpressed in several cancer types and shares similar oncogenic characteristic with *CCND1* in tumor tissues [16,17]. Additionally, cell cycle progression is also regulated by one of the serine/threonine protein kinases: polo-like kinase 1 (*PLK1*) [18]. Recently, the amplification and overexpression of *PLK1* were reported in a variety of tumors [19]. Its role in cancer includes cell differentiation and tumor progression, consequently resulting in poor clinical outcomes and therefore promoting the need for *PLK1* inhibitors in cancer [20,21]. In addition, emerging studies have shown that the suppression of *PLK1* by current inhibitors results in the downregulation of *CCND1* overexpression in tumors [22,23]. Additionally, the overexpression of *PLK1* in tumor cells was shown to activate cancer stem cells (CSCs), with the cluster of differentiation 44 (*CD44*) surface marker mostly being expressed [24].

Comprehensive computational software has been utilized most recently in drug target discovery, by applying in silico bioinformatics predictions for drug targets, as well as screening cancer cells at the gene level in a search for novel drug targets [25]. Moreover, the use of available online databases has improved data analysis and drug target identification [26]. NSC765600 and NSC765691 are both small molecules, respectively derived from diflunisal (Pubchem CID: 3059) and fostamatinib (Pubchem CID: 11671467) [27,28], which were chemically synthesized in our lab [29], where we produced open-ring (NSC765600) and closed-ring (NSC765691) derivatives. In this study, we applied drug target predictions and identified *CCND1/CDK4/PLK1/CD44* as potential drug candidates of NSC765600 and NSC76569, and further performed molecular docking, which revealed putative binding energies of *CCND1/CDK4/PLK1/CD44* with NSC765600 and NSC76569. Furthermore, we showed that both NSC765600 and NSC76569 displayed antiproliferative and cytotoxic activities in vitro against a panel of NCI-60 human cancer cell lines.

## 2. Materials and Methods

### 2.1. Pharmacokinetic (PK), Drug-Likeness, and Medicinal Chemical Analyses

To analyze the drug-likeness, medicinal chemistry, PKs, and the adsorption, distribution, metabolism, excretion, and toxicity (ADMET) properties of our compounds (NSC765600 and NSC765691), we utilized the SwissADME software developed by the Swiss Institute of Bioinformatics (http://www.swissadme.ch, 27 February 2021) [30]. Drug-likeness properties were analyzed according to the Lipinski (Pfizer) rule-of-five, Ghose (Amgen), Veber (GSK), Egan (Pharmacia), and Veber (GSK); in addition, the Lipinski (Pfizer) rule-of-five defines four simple physicochemical parameters with the following ranges (molecular weight ≤ 500, log *p* ≤ 5, hydrogen bond donors ≤ 5, and hydrogen bond acceptors ≤ 10) [31] for drug-likeness and drug discovery. A bioavailability score was calculated based on the probability of the compound having at least 0.1 (10%) oral bioavailability in rats or measurable Caco-2 permeability [32], while gastrointestinal absorption (GIA) and brain penetration properties were analyzed using the brain or intestinal estimated permeation (BOILED-Egg) model [33].

### 2.2. Identifying the Molecular Targets of NSC765600 and NSC765691

Target genes of NSC765600 and NSC765691 were predicted using the Swiss target prediction online tool (http://www.swisstargetprediction.ch/, 27 February 2021), which uses the principle of similarity to predict the drug target. The predictions are also based on “probability”, which is derived from the target score to assess the likelihood of the predicted targets being correct, and the values are based on the correspondence of the average precision (i.e., number of true-positives divided by the total number of predicted targets at different thresholds). These predictions were then used in in vitro and in vivo experiments [34]. Furthermore, we used the DTP-COMPARE algorithms [35] to identify the fingerprint (activity patterns) of NSC765600 and NSC765691, which was based on the correlation of the compounds with NCI synthetic compounds and standard agents. Herein, we used the NSC IDs of both compounds as “seed for the COMPARE algorithms”.

### 2.3. Bioinformatics Predictions

The micro-RNA (mRNA) levels of differentially expressed genes (DEGs) of *CCND1*, *CDK4*, *PLK1*, and *CD44* in tumorous versus normal tissues from various cancers in The Cancer Genome Atlas (TCGA) database were analyzed using UALCAN (http://ualcan.path.uab.edu/, 28 March 2021) and GEPIA2 (http://gepia2.cancer-pku.cn/#index, 28 March 2021 bioinformatics software with default settings.

### 2.4. Protein-Protein Interaction (PPI) Analysis

In a further analysis, we applied the GeneMANIA (https://genemania.org/, 22 February 2021) and STRING (https://string-db.org/, 22 February 2021) online analytical tools to predict PPIs between expressed genes, and a gene ontology (GO) enrichment analysis including biological processes and Kyoto Encyclopedia of Genes and Genomes (KEGG) pathways. The results displayed in a network for PPIs indicated correlations of *CDK4* with *CCND1*, *CDK4* with *CD44*, *CCND1* with *CD44*, *CDK4* with *PLK1*, and *PLK1* with *CCND1*, wherein nodes symbolize genes and edges represents networks.

### 2.5. Analysis of Genomic Alterations of CCND1/CDK4/PLK1/CD44 in Multiple Cancer Types

To analyze genomic alterations of *CCND1/CDK4/PLK1/CD44* gene signatures, we applied the Oncoprint feature of cBioportal software (https://www.cbioportal.org/, 14 March 2021), which categorizes gene alterations based on percentages of individual genes due to amplification, and in a further analysis, we determined the alteration frequencies of our gene signatures in multiple cancer types. Next, we used a mutually exclusivity panel analysis, a sub-tool of cBioportal software, to determine if the remaining altered genes within the entire set co-occurred with *CCND1/CDK4/PLK1/CD44* signatures, and the threshold was designed with *p* < 0.001.

### 2.6. In Silico Analysis of Molecular Docking of Receptors and Ligands

The potential inhibitory activities of NSC765600 and NSC765691 were analyzed by the in silico molecular docking of oncogene signaling of *CCND1*, *CDK4*, *PLK1*, and *CD44*, compared to the standard inhibitors fascaplysin, ribociclib, and volasertib of *CCND1*, *CDK4*, and *PLK1*, respectively. The 3D structures of NSC765600 and NSC765691 were drawn in sybyl mol2 using the Avogadro molecular builder and visualization tool (http://avogadro.openmolecules.net, 8 February 2021) [36]. The 3D structural conformers of fascaplysin (CID: 7329), ribociclib (CID: 4431912), and volasertib (CID: 10461508) were downloaded in SDF file format from the PubChem database. The mol2 and SNF format structures were subsequently converted to PDB format using the Pymol molecular visualization system (https://pymol.org/2/, 8 February 2021), and later to the PDBQT file format using autodock tool software (http://autodock.scripps.edu/resources/adt, 8 February 2021). Crystal structures of *CCND1* (PDB-6P8G), *CDK4* (PDB-4O9W), *PLK1* (PDB-2W9F), and *CD44* (PDB-1UUH) were retrieved from the Protein Data Bank (PDB) (https://www.rcsb.org/, 8 February 2021), in PDB format, and eventually converted to PDBQT files with the autodock tool. Molecular docking was performed using autodock tools [37]. For visualization, we used pymol, and further interactive 3D visualization and interpretation were analyzed using the BIOVIA discovery studio tool [38].

### 2.7. In Vitro Screening of NSC765600 and NSC765691 against Full National Cancer Institute (NC I)-60 Cell Panels of Human Tumor Cell Lines

NSC765600 and NSC765691 were submitted for screening to the NCI (Rockville, MD 20850, USA) to screen its panel of NCI-60 cancer cell lines. The two compounds were screened for antiproliferative and cytotoxic effects at an initial single dose (10 μM) against the full NCI-60 cell line panel, which includes leukemia, non-small cell lung cancer (NSCLC), melanomas, renal cancer, breast cancer, central nervous system (CNS) cancers, ovarian cancer, and prostate cancer, in agreement with the protocol of the NCI (http://dtp.nci.nih.gov, 1 January 2021).

### 2.8. Data Analysis

Pearson’s correlations were used to assess correlations between *CCND1*/*CDK4*/*PLK1*/*CD44* expressions in multiple cancer types. The statistical significance of DEGs was evaluated using the Wilcoxon test. * *p* < 0.05 was accepted as being statistically significant. The Kaplan–Meier curve was employed to present patient survival in different cancer cohorts.

## 3. Results

### 3.1. NSC765600 and NSC765691 Adhere to the Required Drug-Likeness Criteria

NSC765600 (open ring) (Figure 1a) [27,39] and NSC765691 (closed ring) (Figure 1b) are respective fostamatinib- and salicylanilide-derived compounds [40]. Using a computer simulation, we applied the criteria described for drug-likeness concepts to identify the potential of these novel compounds for drug discovery and development [41]. The GIA and blood–brain barrier (BBB) permeability of these small compounds were evaluated using the BOILED-Egg predictive model [33]. Both NSC765600 and NSC765691 exhibited prospective GIA and BBB permeability. Six important properties for oral bioavailability, which enabled us to assess the drug-likeness of NSC765600 and NSC765691, were evaluated as displayed on the bioavailability radar (Appendix A). The compounds were evaluated according to their molecular weight, polarity, solubility, saturation, flexibility, and lipophilicity, and both met drug-likeness requirements. Interestingly, NSC765600 and NSC765691 showed favorable PKs, drug-likeness, and medicinal chemical properties, and met the Lipinski’s rule-of-five for drug-likeness and drug discovery, with good synthetic accessibilities of 2.64 and 3.48, respectively (Table 1 and Table 2). The bioavailability of the compounds based on GIA revealed a score of 0.55 (55%) for both compounds, which indicates acceptable PK properties. Since both compounds met the drug-likeness criteria, we further investigated the activity patterns (fingerprints) and mechanistic relations of NSC765600 and NSC765691 with NCI synthetic compounds and standard anticancer agents, as stipulated by the development therapeutics program (DTP). This program is used to identify the molecular targets and mechanisms of an unknown compound from an available known compound in the NCI databases [42]. After a comparative analysis, as anticipated, NSC765600 and NSC765691 shared similarities fingerprints and mechanisms with NCI synthetic compounds and standard anticancer agents, with Pearson’s correlations of *p* = 0.46~0.25 for NSC765600 and *p* = 0.5~0.4 for NSC76569 (Table 3).

### 3.2. CCND1/CDK4/PLK1/CD44 Gene Signatures are Potential Drug Targets for NSC765600 and NSC765691

We applied the Swiss target prediction tool to NSC765600 and NSC765691 and identified potential druggable genes, including *MTOR*, *PIK3R1*, *GSK3A*, *PIK3CD*, *PLK1*, *CCND1*, *CDK2*, and *CDK4* for NSC765600 and *PLK1*, *CCND1*, *TLR9*, *GSK3B*, *PTK2*, and *CDKs*, among other target genes, for NSC765691. There are various target classes that the prediction tool identified for both compounds, such as kinases, enzymes, family A G protein-coupled receptors, proteases, phosphodiesterase, isomerase, oxidoreductase, and Toll-like and interleukin (IL)-1 receptors. The calculation results from the software were based on the “probability values” derived from the target score to assess the likelihood of the predictions being correct. Herein the probability showed a similar value of 0.11 across all predicted target genes. (Appendix A, Table 4 and Table 5).

### 3.3. CCND1/CDK4/PLK1/CD44 are Overexpressed in Multiple Cancers and Associated with Poor Prognoses

An analysis using the UALCAN (http://ualcan.path.uab.edu/, 28 March 2021) online bioinformatics tool with default settings showed that the mRNA levels of *CCND1*/*CDK4*/*PLK1*/*CD44* were higher in tumor tissues, compared to normal tissues, in multiple cancer types (Figure 2). To further investigate the correlations between *CCND1*/*CDK4*/*PLK1*/*CD44* siganature expressions and clinical prognoses, we predicted the overall survival percentage using multiple datasets, which is a sub-tool of survival analysis on the *GEPIA2* software (http://gepia2.cancer-pku.cn/, 28 March 2021). Based on our analysis, we found that *CCND1*/*CDK4*/*PLK1*/*CD44* signatures were linked with shorter survival percentages in a group of multiple cancer types, including bladder carcinoma, breast cancer, colon adenocarcinoma, glioblastoma multiforme, head and neck cancers, leukemia, *NSCLC*, and ovarian cancer (Figure 3).

Moreover, we explored the GEPIA tool and Human Protein Atlas (HPA) database for immunohistochemistry (IHC) (HPA; www.proteinatlas.org/, 27 March 2021), to compare the expression levels of *CCND1*, *PLK1*, and *CD44* between tumor samples and normal samples. HPA is a public database that displays IHC human protein expressions for tumor tissue and normal tissues [43]. Based on our analysis results, *CCND1*, *PLK1*, and *CD44* genes’ signaling were significantly expressed in multiple cancer types, including melanoma (SKMC), lung cancer (LAUD), ovarian cancer (OV), renal cancer (READ), bladder cancer (BLCA), colorectal cancer (COAD), prostate cancer (PRAD) and breast cancer (BRCA). In addition, the HPA data of *CCND1*, *PLK1*, and *CD44* expression in tumor smaples exhibited high staining intensities, and all the IHC images were obtained from the HPA database to validate the expression of genes in the cancer types at the protein level. Staining intensity was analyzed based on low staining, medium staining, and high staining, and *p <* 0.05 was considered statistically significant (Figure 4).

For futher analysis, we applied the GeneMANIA (https://genemania.org/, 22 February 2021) and STRING databases (https://string-db.org/cgi/, 22 February 2021) to predict PPIs between all four oncogenes. After considering the gene neighborhood, gene fusion, gene co-occurence, and coexpression, as anticipated, interactions were detected between *CDK4* with *CCND1*, *CDK4* with *CD44*, *CDK4* with *PLK1*, and *CCND1* with *PLK1* within network clustering. The numbers of nodes and edges were ultimately increased to 24 and 192, respectively, within the network, with an interation enrichment average local clustering coefficient of 0.845 and *p* < 1.0 × 10^−16^ (Figure 5A,B The accompanying table shows all other interacting proteins with *CCND1/CDK4/PLK1/CD44*, and the confidence cutoff value representing the interaction links was adjusted to 0.900 as the highest scoring link. In addition, we performed a gene enrichment analysis and predicted the major biological processes associated with the *CCND1*/*CDK4*/*PLK1*/*CD44* gene signature. The correlated gene clusters were cellular protein modification processes, regeneration, DNA damage checkpoints, and transition of the mitotic cell cycle (Figure 5C). Moreover, *CCND1*/*CDK4*/*PLK1*/*CD44* oncogenic signaling-correlated gene clusters affected 10 pathways, which were significantly associated with serveral functions, as shown in the KEGG pathways (Figure 5D. Pathways in cancer, the *PI3K-AKT* signaling pathway, and prostate cancer were the most significant pathways associated with *CCND1*/*CDK4*/*PLK1*/*CD44* siganling. Additionally, the seven other involved pathways included gliomas, NSCLC, gastric cancer, herpesviruses, hepatocellar carcinoma, breast cancer, and bladder cancer.

### 3.4. CCND1/CDK4/PLK1/CD44 Genes are Altered in Multiple Cancer Types

We applied the Oncoprint feature of cBioportal software (https://www.cbioportal.org/, 14 March 2021), which categorized gene alterations of *CCND1*/*CDK4*/*PLK1*/*CD44* based on percentages of separate genes due to amplification. The results of the analysis were as follows: 6% for *CCND1*, 2.8% for *CDK4*, 1.7% for *PLK1*, and 1.6% for *CD44* in multiple cancers, which included missense mutations (green), amplifications (red), deep deletions (blue), and no alterations (gray) (Figure 6A). In a further analysis, we determined the alteration frequencies of *CCND1*/*CDK4*/*PLK1*/*CD44* gene signatures in multiple cancer types, and we then used a mutually exclusivity panel analysis, which is a sub-tool of the cBioportal software, and found the rest of the altered genes within the entire set that co-occurred with *CCND1*/*CDK4*/*PLK1*/*CD44* signatures, at a threshold with *p* < 0.001 as significant (Figure 6B–E, and accompanying table).

### 3.5. Determining Protein–Ligand Interactions (PLIs)

We applied molecular docking simulations, which demonstrated distinct binding abilities of NSC765600 and NSC765691 with *CCND1*/*CDK4*/*PLK1*/*CD44* oncogenic signaling pathways. The two compounds achieved good binding affinities with the target proteins. From our docking analysis, the estimated binding energies (ΔG) of NSC765600 with *CCND1* (PDB:6P8G), *CDK4* (PDB:4O9W), *PLK1* (PDB:2W9F), and *CD44* (PDB-1UUH) complexes were −9.3, −8.0, −7.4, and −7.0 kcal/mol, respectively (Figure 7, and accompanying table). Interestingly, NSC765691 displayed even greater binding free energies of −9.6, −8.0, −7.7, and −7.3 kcal/mol with *CCND1, CDK4, PLK1*, and *CD44,* respectively (Figure 8, and accompanying table). The docking analysis results have been visualized using Discovery studio, and the analysis has revealed the interactions of five conventional hydrogen bonds and their minimal distance constraints, including ASN198 (2.51 Å) and ASN198 (2.04 Å) with *CCND1*, ARG87 (2.29 Å) with *CDK4,* SER439 (2.70 Å) with *PLK1,* and CRY77 (2.90 Å) with *CD44*, all in complex with NSC765600.

Further supporting interactions were revealed, with their designated amino acids, including carbon–hydrogen bonds (PRO200), Pi-anion (GLU35), Pi-sigma (ALA39), and Pi-alkyl (MET82, ALA190, PRO157, PRO40, and PRO199) for the *CCND1*-NSC765600 complex; carbon–hydrogen bonds (GLU64, THR37), Pi-sigma (LEU91), and Pi-alkyl (LYS149 and ALA65) for the *CDK4*-NSC765600 complex; carbon–hydrogen bonds (ASN437 and LEU435), Pi-anion (ARG594), Pi-sigma (THR513), and Pi-alkyl (TRP514 and ARG512) for the *PLK1*-NSC765600 complex; and Pi-alkyl (ILE91 and ILE96) for the *CD44*-NSC765600 complex (Figure 7, and accompanying table). Moreover, the visualization analysis showed even more interactions of eight conventional hydrogen bonds and their minimal distance constraints, including ARG89 (2.51 Å), ASN151 (2.04 Å), LYS149 (1.85 Å), and SER (2.89 Å) with *CCND1*; ARG61 (1.76 Å) with *CDK4*; LEU491 (1.90 Å) and TRP414 (2.02 Å) with *PLK1*; and CYS28 (2.09 Å) with *CD44*, all in complex with NSC76591. Interactions were also further stabilized by other interactions with their assigned amino acids, including Pi-alkyl (LEU490 and VAL415) for the *CCND1*-NSC765691 complex; carbon–hydrogen bonds (GLU64, GLU67, and SER90), and Pi-alkyl (LYS149 and LEU91) for the *CDK4*-NSC765691 complex; Pi-alkyl (LEU490 and VAL415) for the *PLK1*-NSC765691 complex; and carbon–hydrogen bonds (VAL148, ILE26, and GLU75) and Pi-alkyl (ARG150 and HIS35) for the *CD44*-NSC765600 complex (Figure 7 and Figure 8, and accompanying tables). For further analysis, we compared the docking analysis results of NSC765600 and NSC765691 with standard inhibitors of fascaplysin, ribociclib, and volasertib for *CCND1*, *CDK4*, and *PLK1,* respectively. Interestingly, the standard inhibitors displayed the lowest binding free energies of −7.5 and −7.6 kcal/mol for fascaplysin and ribociclib, respectively, but with the exception of volasertib, which showed a higher binding energy of 7.9 kcal/mol, compared to our compounds (Figure 9). Therefore, the simulations predicted NSC765600 and NSC765691 to be potential multi-target inhibitors with high confidence.

### 3.6. NSC765600 and NSC765691 Display Antiproliferative and Cytotoxic Effects against a Panel of 60 Human Tumor Cell Lines

The anticancer activities of NSC765600 and NSC765691 were evaluated against a panel of 60 human cancer cell lines available from the US NCI-developed therapeutic program. This analysis included the antiproliferative and cytotoxic activities of the compounds on melanomas, central nevous system (CNS) cancers, renal cancer, breast cancer, NSCLC, leukemia, colon cancer, prostate cancer, and ovarian cancer [35]. The analytical results revealed the antiproliferative and cytotoxic effects of NSC765600 and NSC765691 against all cancer cell lines present in NCI data. The compunds exhibited anticancer activities after an initial dose of 10 μM, and the treatment efficacy was represented by the percentage (%) growth (Figure 10).

## 4. Discussion

Cancer progression has evolved over the years, and most fatalities now occur due to metastasis and resistance to therapeutic interventions [44]. Despite current advanced treatment interventions, including radiation, chemotherapy, and surgery, the median overall survival in patients with advanced disease is still under 5 years in the majority of cancers [5]. This therefore indicates the urgent need to develop novel and improved therapeutics that can be used either in combination with chemotherapy or as single agents. Recently, a vast amount of attention has been focused on small molecules as targeted treatments for cancer [42]. In this study, we revealed the inhibitory activities of two novel compounds, NSC765600 and NSC765691, in multiple cancers. Precepts of drug-likeness allowed us to identify druggable targets in the initial stage of drug discovery and development [45]. Therefore, we utilized the SwissADME software developed by the Swiss Institute of Bioinformatics [30]; interestingly, the GIA and BBB permeability of these two small compounds exhibited good prospects. In addition, the compounds passed drug-likeness requirements, and were evaluated according to their molecular weight, polarity, solubility, saturation, flexibility, and lipophilicity. Moreover, NSC765600 and NSC765691 showed favorable PK, drug-likeness, and medicinal chemical properties, and met Lipinski’s rule-of-five for drug-likeness and drug discovery, with good synthetic accessibilities of 2.64 and 3.48, respectively (Table 1 and Table 2). The bioavailability of the compounds based on GIA indicated a score of 0.55 (55%) for both compounds, which indicates acceptable PK properties, implying that both NSC765600 and NSC765691 are drug-like compounds.

Since both compounds met the drug-likeness criteria, we further investigated the fingerprints and mechanistic relations of NSC765600 and NSC765691 with NCI synthetic compounds and standard anticancer agents, as stipulated by the DTP [42]. After a comparative analysis, as anticipated, NSC765600 and NSC765691 shared similar fingerprints and mechanisms with NCI synthetic compounds and standard anticancer agents, with Pearson’s correlations of *p* = 0.46~0.25 and *p* = 0.4~0.5 for NSC765600 and NSC76569, respectively (Table 3). To further evaluate the biological and inhibitory effects of NSC765600 and NSC765691 in cancer, we identified *CCND1*/*CDK4*/*PLK1*/*CD44* as potential druggable candidates for both compounds by using online prediction tools. Various target classes were also identified, among which were kinases and enzymes, suggesting that NSC765600 and NSC765691 are satisfactory compounds to be used as targets for *CCND1*/*CDK4*/*PLK1*/*CD44* oncogenic signatures. The predictions were also based on “probability value”, which is derived from the target score to assess the likelihood of the predicted targets being correct. The values were based on the correspondence of the average precision (i.e., number of true-positives divided by the total number of predicted targets at different thresholds) [34]. Herein, the results showed similar probability values of 0.11 across all target genes predicted to be correct. These predictions will be further used in in vitro and in vivo experiments.

In silico molecular docking is a computer simulation tool, which has been applied in drug design and development over the years, with the aim of predicting ligand–protein binding interactions [46]. Herein, we applied docking simulations of NSC765600 and NSC765691 with *CCND1*/*CDK4*/*PLK1*/*CD44* oncogenic signaling pathways, and the results showed the unique interactions of NSC765600 and NSC765691 with the *CCND1*/*CDK4*/*PLK1*/*CD44* complex, and also exhibited predominant binding free energies for both compounds. Next, we compared the binding affinities of the NSC765600 and NSC765691 standard inhibitors, fascaplysin, ribociclib, and volasertib, with *CCND1*, *CDK4*, and *PLK1,* respectively. From the analysis, NSC765600 and NSC765691-*CCND1*, *CDK4*, *PLK1*, and *CD44* complexes displayed the highest binding energies of −9.3, −8.3, −7.4 and −7.0 kcal/mol and (−9.6, −8.0, −7.7 and −7.3 kcal/mol, respectively (Figure 7 and Figure 8 and accompanying tables), as compared to the standard inhibitors’ results, which were −7.5, −7.6 and −7.9 kcal/mol for fascaplysin, ribociclib, and volasertib-*CCND1*, *CDK4*, and *PLK1,* respectively (Figure 9, accompanying table). Interestingly, NSC765691 (closed ring) showed even stronger binding affinities with the *CCND1*/*CDK4*/*PLK1*/*CD44* oncogenes, compared to NSC765600 and selected standard inhibitors of *CCND1*/*CDK4*/*PLK1*. This confirms the findings from previous studies, which showed that closed-ring structures are more stable in drug design and development, compared to open-ring structures [47].

When we further applied a bioinformatics analysis using UALCAN and GEPIA2, we identified increased mRNA expression levels of *CCND1/CDK4/PLK1/CD44* in multiple cancer types, which resulted in shorter overall survival times and poor prognoses in cancer patients, with p-values less than 0.05 and hazard ratios more than 1 considered to be significant (Figure 2 and Figure 3). Additionally, PPIs showed interactions according to the gene neighborhood, gene fusion, gene co-occurance, and coexpression of *CDK4* with *CCND1*, *CDK4* with *CD44*, *CDK4* with *PLK1*, and *CCND1* with *PLK1*, with an enrichment average local clustering coefficient of 0.845 and *p* < 1.0 × 10^−16^. GO enrichment revealed major biological processes and pathways associated with *CCND1/CDK4/PLK1/CD44* in different cancers (Figure 5). We further applied the cBioportal software and found that *CCND1/CDK4/PLK1/CD44* genes were altered and co-occurring with other different genes in multiple cancers. The analysis showed the amplification of *CCND1/CDK4/PLK1/CD44* based on percentages of separate genes, with 7% for *CCND1*, 2.9% for *CDK4*, 1.7% for *PLK1*, and 1.8% for *CD44* in multiple cancers, including the missense mutation amplifications, deep deletions, and alteration frequencies of these oncogenes.

Based on these finding, NSC765600 and NSC765691 exhibited potential inhibitory effects on the *CCND1/CDK4/PLK1/CD44* oncogenic pathway, and passed all the required criteria for drug-likeness precepts as novel compounds. Further in vitro and in vivo studies in tumor-bearing mice will be performed to investigate the potential treatment efficacies of the novel NSC765600 and NSC765691 small molecules.

## 5. Conclusions

In conclusion, we have shown that *CCND1*/*CDK4*/*PLK1*/*CD44* are potential drug targets for NSC765600 and NSC765691 small molecules, and the docking analytical results revealed putative binding energies between the two compounds and *CCND1*/*CDK4*/*PLK1*/*CD44* oncogenic signatures. We also showed the antiproliferative and cytotoxic effects of NSC7565600 and NSC765691 against a panel of NCI-60 cancer cell lines. This suggests the potential of NSC765600 and NSC765691 to inhibit *CCND1*/*CDK4*/*PLK1*/*CD44* expressions in cancer. Further in vitro and in vivo studies in tumor-bearing mice will be performed to investigate the potential treatment efficacies of the novel NSC765600 and NSC765691 small molecules.

## Figures and Tables

**Figure 1 cancers-13-02523-f001:**
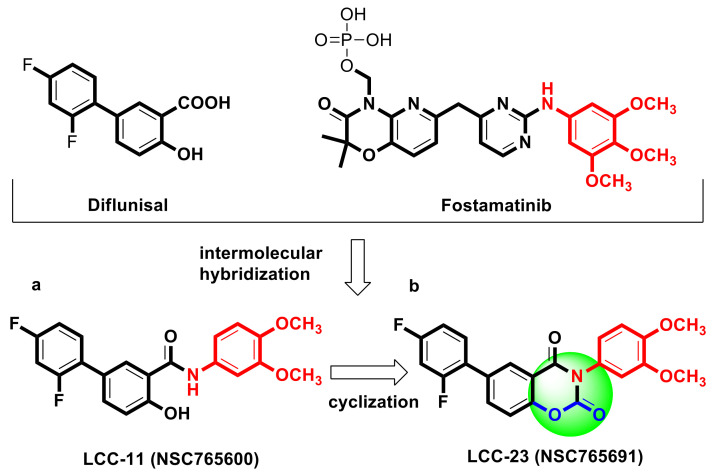
NSC765600 (**a**) and NSC765691 (**b**) are both small molecules derived from diflunisal (Pubchem CID: 3059) and fostamatinib (Pubchem CID: 11671467), respectively.

**Figure 2 cancers-13-02523-f002:**
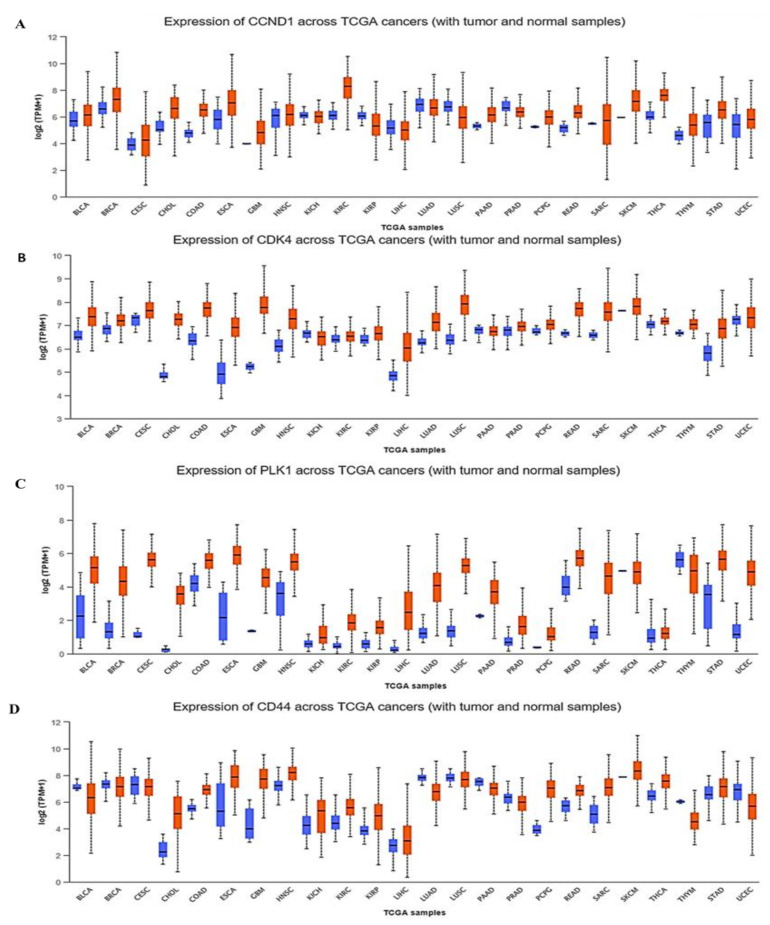
Cyclin D1 (CCND1)/cyclin-dependent kinase 4 (CDK4)/polo-like kinase 1 (PLK1)/cluster of differentiation 44 (*CD44*) genes are highly expressed in multiple cancer types. TCGA dataset of differential expression levels of (**A**) CCND1, (**B**) CDK4, (**C**) PLK1, and (**D**) *CD44*, based on patients with tumor tissues (red) and normal tissues (blue).

**Figure 3 cancers-13-02523-f003:**
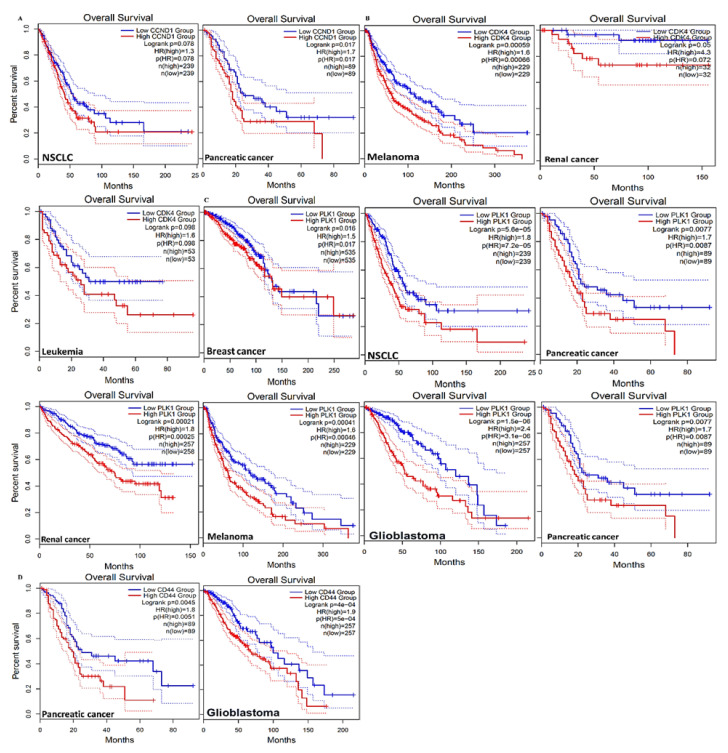
Predictions of overall survival percentages using *CCND1*/*CDK4*/*PLK1*/*CD44*. Increased mRNA levels of *CCND1*, *CDK4*, *PLK1* and *CD44* oncogenic signaling were found to be associated with shorter survival percentages in the following selected cancer types: NSCLC, pancreatic cancer, melanoma, renal cancer, leukemia, breast cancer and glioblastoma multiforme. The mRNA levels of (**A)** *CCND1* (**B**) *CDK4* (**C**) *PLK1* and (**D**) *CD44* were shown to be associated with short survival percentages in the abovementioned cancer types, with hazard ratio > 1 and *p* < 0.05 considered to be significant.

**Figure 4 cancers-13-02523-f004:**
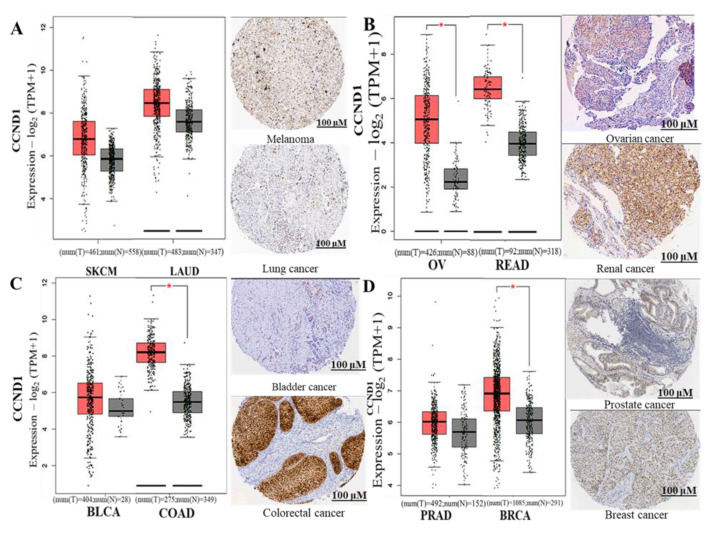
Validation of the expression of *CCND1*, *CDK4* and *PLK1* gene signtures on protein level using GEPIA and HPA databases. (**A**–**D**) CCND1, (E-H) PLK1, (I-L) CD44 GEPIA analysis expressed genes. HPA staining intensity was analyzed based on low staining, medium staining, and high staining. (**A**–**E**) CCND1 dispalyed low staining intensities in ovarian cancer and colorectal cancer, medium staining on breast cancer and prostate cancer, and high staining intensities on lung cancer, melanoma, renal cancer and bladder cancer. (**E**–**H**) PLK1 exhibited medium staining intensities on colorectal cancer, lung cancer, renal cancer, balddder cancer and prostate cancer, while high staining was observed on melanoma, ovarian cancer and breast cancera. (**I**–**L**) CD44 showed high staining intensities on all the abovementioned cancer types. Color images are available online. *CDK4* was not included since the data have not yet been made available on the HPA database. The staining quality was <75%, and ** p <* 0.05 was considered statistically significant, All images can be found online.

**Figure 5 cancers-13-02523-f005:**
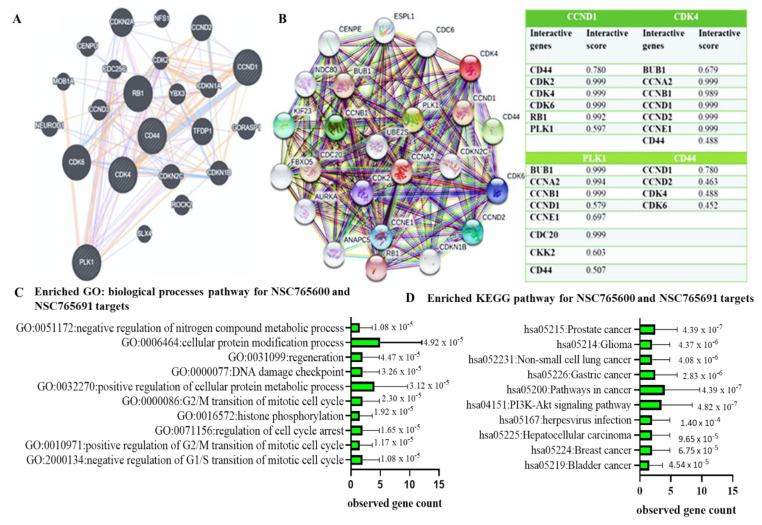
Protein–protein interaction (PPI) network among *CCND1/CDK4/PLK1/CD44* genes in the GeneMANIA and STRING datasets. Interactions are shown after considering the gene neighborhood, gene fusion, gene co-occurance, and coexpression of *CDK4* with *CCND1*, *CDK4* with *CD44*, *CDK4* with *PLK1*, and *CCND1* with *PLK1*. (**A**) PPI network in the GeneMANIA dataset. (**B**) PPI network in the STRING dataset with network clustering. The numbers of nodes and edges were ultimately increased to 24 and 192, respectively, within the network, with an interation enrichment average local clustering coefficient of 0.845 and *p* < 1.0 × 10^−16^. The accompanying table shows all other interacting proteins with *CCND1/CDK4/PLK1/CD44*, and the confidence cutoff value representing interaction links was adjusted to 0.900 as the highest scoring link. (**C**) Major biological processes associated with *CCND1/CDK4/PLK1/CD44* oncogenic signature-correlated gene clusters. (**D**) KEGG pathway showing 10 affected pathways with *CCND1/CDK4/PLK1/CD44* signatures, which were significantly associated with several functions. *p*-values are indicated in each panel.

**Figure 6 cancers-13-02523-f006:**
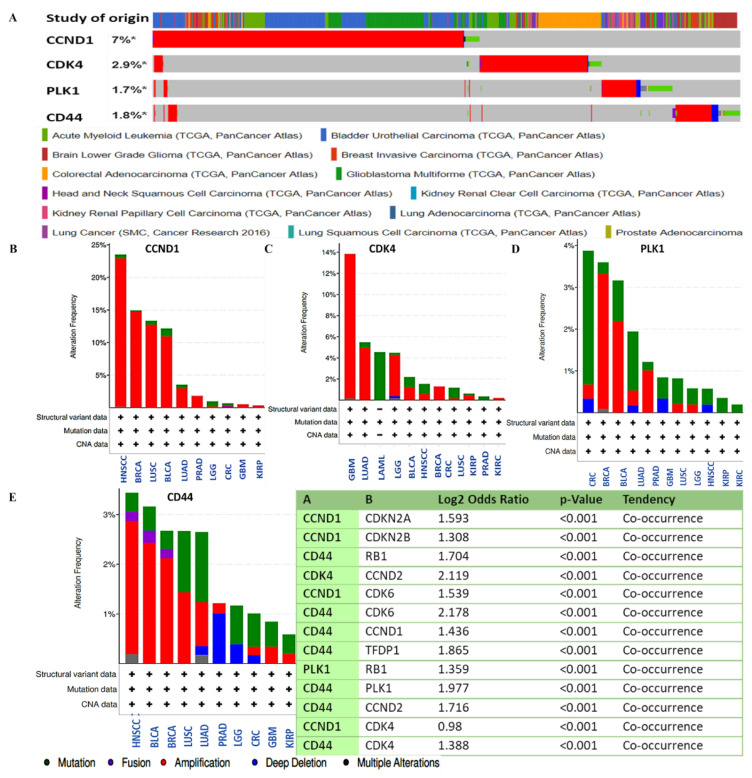
*CCND1/CDK4/PLK1/CD44* oncogenes were amplified and found to co-occur in multiple cancer types. (**A**) Oncoprint analysis showed amplification (marked with *) of *CCND1/CDK4/PLK1/CD44* based on percentages of separate genes, with 7% for *CCND1*, 2.9% for *CDK4*, 1.7% for *PLK1*, and 1.8% for *CD44* in multiple cancers, including missense mutations (green), amplifications (red), deep deletions (blue), or no alterations (gray). (**B**–**E**) Alteration frequencies of *CCND1/CDK4/PLK1/CD44* signaling pathways in multiple cancer types. Accompanying table: co-occurrence of *CCND1/CDK4/PLK1/CD44* signatures with other genes within the set, at a threshold of *p* < 0.001 as significant.

**Figure 7 cancers-13-02523-f007:**
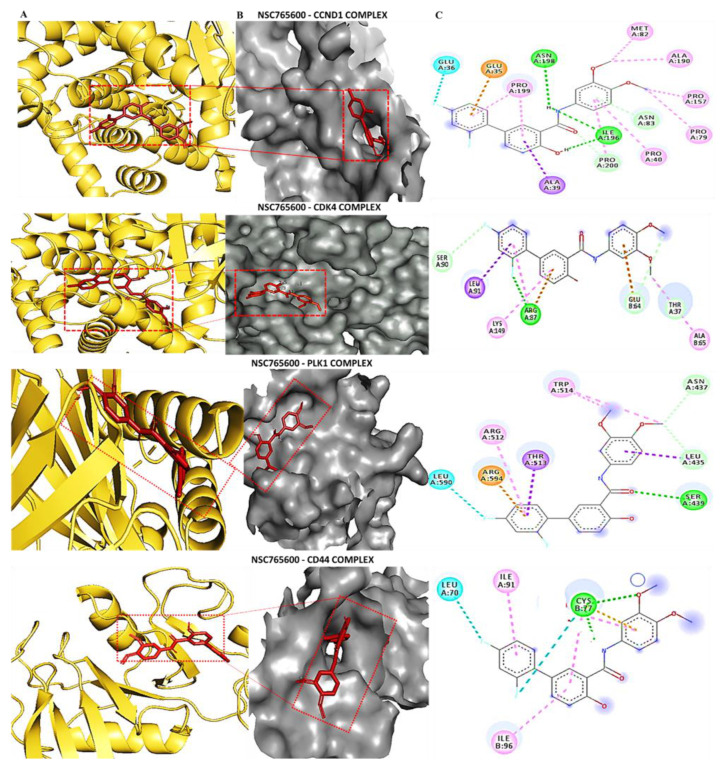
Docking profiles of NSC765600 with *CCND1*, *CDK4*, *PLK1*, and *CD44*. (**A**) Ligand–receptor interactions between *CCND1/CDK4/PLK1/CD44* and NSC765600 in a two-dimensional representation. (**B**) Binding pocket presentation of the NSC7656000-*CCND1/CDK4/PLK1/CD44* complex. (**C**) Visualization of putative docking poses of ligand–receptor interactions displayed by conventional hydrogen bonds. The accompanying table gives the binding energies of ligand–receptor interactions, including different types of interactions and the amino acid residues involved.

**Figure 8 cancers-13-02523-f008:**
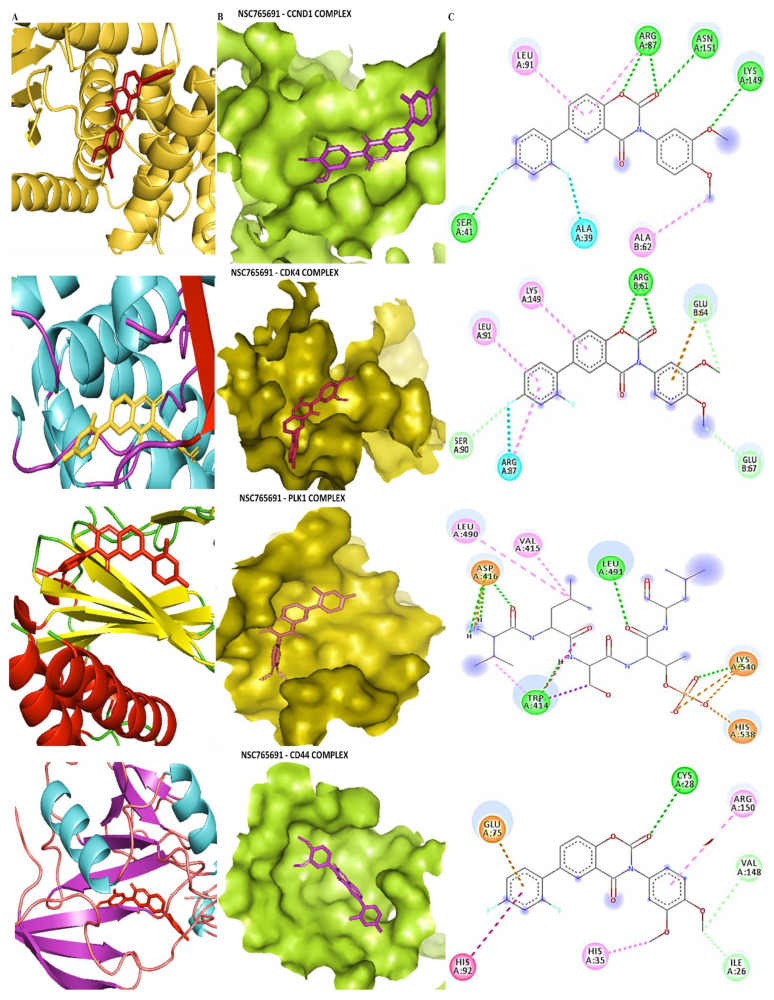
Docking profiles of NSC765691 with *CCND1*, *CDK4*, *PLK1*, and *CD44*. (**A**) Ligand–receptor interactions between *CCND1/CDK4/PLK1/CD44* and NSC765691 in a two-dimensional representation. (**B**) Binding pocket presentation of the NSC7656000-*CCND1/CDK4/PLK1/CD44* complex. (**C**) Visualization of putative docking poses of ligand–receptor interactions displayed by conventional hydrogen bonds. The accompanying table gives binding energies of ligand–receptor interactions, including different types of interactions and the amino acid residues involved.

**Figure 9 cancers-13-02523-f009:**
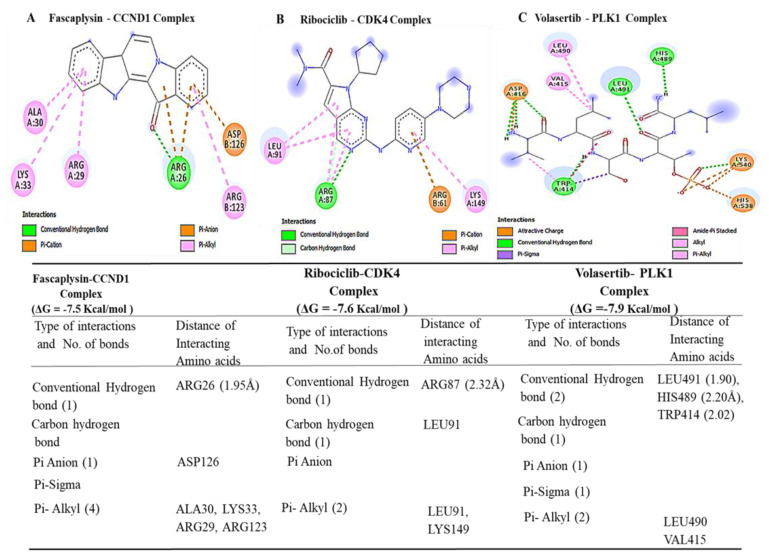
Docking profiles of the standard inhibitors fascaplysin, ribociclib, and volasertib with *CCND1, CDK4,* and *PLK1,* respectively. Visualization of the highest docking poses of ligand–receptor interactions displayed by conventional hydrogen bonds for the fascaplysin–*CCND1* complex (**A**), ribociclib–*CDK4* complex (**B**), and volasertib–PLK1 complex (**C**). the standard inhibitors displayed the lowest binding free energies of −7.5 and −7.6 kcal/mol for fascaplysin and ribociclib, respectively, but with the exception of volasertib, which showed a higher binding energy of 7.9 kcal/mol, compared to our compounds. The accompanying table gives the binding energies of ligand–receptor interactions, including different types of interactions and the amino acid residues involved.

**Figure 10 cancers-13-02523-f010:**
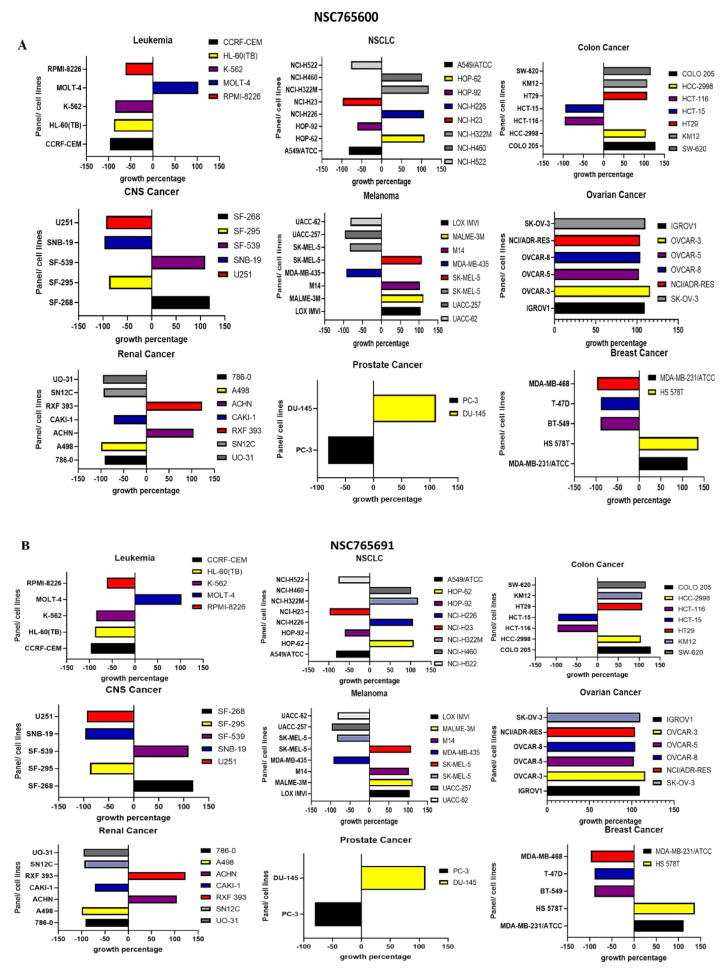
NSC765600 and NSC765691 exhibit anticancer activities against a panel of NCI-60 cancer cell lines. The sensitivities of the NCI-60 cancer cell lines were evaluated using (**A**) NSC765600 and (**B**) NSC765691 at an initial dose of 10 μM. The zero on the *x*-axis indicates the mean percentage of treated tumor cell lines. The anticancer effects of the compounds on the cell lines are represented by growth percentage (left) relative to mean growth percentage (right). When the growth percentage values are below zero (0), this indicates cell cytotoxicity/cell death, while when the mean growth percentage values are above 0, this indicates antiproliferative effects.

**Table 1 cancers-13-02523-t001:** Physiochemical properties, pharmacokinetics, drug-likeness, and medical chemistry of NSC765600.

Physicochemical Properties Based on Bioavailability Radar of NSC765600		Recommended Value
Formula	C_21_H_17_F_2_NO_4_	
Molecular weight	385.36 g/mol	150–500 g/mol
Fraction Csp3	0.10	≤1
Number of rotatable bonds	6	≤10
Number of H-bond acceptors	6	≤12
Number of H-bond donors	2	≤5
Molar refractivity	101.02	
TPSA	67.79 Å^2^	≤140 Å^2^
Log P_o/w_ (XLOGP3)	5	−0.7–5
Log S (ESOL)	−5.89	0–6
**Pharmacokinetics**		
GI absorption	High	
BBB	Low	
**Drug-likeness**		
Lipinski	Yes; 0 violation	
Ghose	Yes	
Veber	Yes	
Egan	Yes	
Muegge	Yes	
Bioavailability Score	0.55 (55%)	
**Medical Chemistry**		
Synthetic accessibility	2.64	1 (easy to make) and 10 (difficult to make)

**Table 2 cancers-13-02523-t002:** Physiochemical properties, pharmacokinetics, drug-likeness, and medical chemistry of NSC765691.

Physicochemical Properties Based on Bioavailability Radar of NSC765691		Recommended Value
Formula	C_22_H_15_F_2_NO_5_	
Molecular weight	411.36 g/mol	150–500 g/mol
Fraction Csp3	0.09	≤1
Number of rotatable bonds	4	≤10
Number of H-bond acceptors	7	≤12
Number of H-bond donors	0	≤5
Molar refractivity	106.42	
TPSA	70.67 Å^2^	≤140 Å^2^
Log P_o/w_ (XLOGP3)	4.53	−0.7–5
Log S (ESOL)	−5.52	0–6
**Pharmacokinetics**		
GI absorption	High	
BBB	Yes (0.215)	
**Drug-likeness**		
Lipinski	Yes; 0 violation	
Ghose	Yes	
Veber	Yes	
Egan	Yes	
Muegge	Yes	
Bioavailability score	0.55 (55%)	
**Medical Chemistry**		
Synthetic accessibility	3.48	1 (easy to make) and 10 (difficult to make)

**Table 3 cancers-13-02523-t003:** NCI synthetic compounds and standard anticancer agents sharing similar anticancer fingerprints and mechanistic correlations with NSC765600 and NSC765691.

	Rank	*p*	CCLC	Target Descriptor	MW	*p*	CCLC	Target Descriptor	MW
**NSC765600 Fingerprint**	1	0.63	49	Dinoterb	240.21	0.46	52	4-ipomeanol	168.19
	2	0.62	50	8-[(4-tert-butylphenoxy)]	342.4	0.41	49	Piperazine	86.14
	3	0.59	53	Masterid	360.5	0.4	44	Amsacrine	393.5
	4	0.59	52	Nitrodan(usan)	296.3	0.4	52	Fluorodopan	249.67
	5	0.58	41	Resorcinol, 4-hexyl-(8ci)	194.27	0.39	42	Mustard	159.08
	6	0.58	52	Azd-4635	315.73	0.37	50	Tamoxifen	371.15
	7	0.57	50	Chimaphilin	186.21	0.35	52	Topotecan	421.4
	8	0.56	53	10074-g5	332.3	0.33	44	Morpholino	86.11
	9	0.56	53	Gsk586581a	381.4	0.32	52	Procarbazine	221.3
	10	0.55	50	Tioconazole (usan)	387.7	0.28	45	Diaziquone	364.35
	**Rank**	**r**	**CCLC**	**Target Descriptor**	**MW**	**r**	**CCLC**	**Target Descriptor**	**MW**
**NSC765691 Fingerprint**	1	0.69	41	Flavanone	298.3	0.5	56	4-ipomeanol	168.19
	2	0.69	54	13668	217.69	0.49	53	Piperazine	86.14
	3	0.68	56	Isomammein	372.5	0.47	55	Tamoxifen	371.5
	4	0.66	57	C.I. 37525	311.8	0.46	56	Flavoneacetic	280.5
	5	0.65	56	10074-g5	332.3	0.44	56	Sulfoximine	64.09
	6	0.64	47	4-(acetyl) amphilectolide	302.4	0.43	45	Mustard	159.08
	7	0.64	53	Chimaphilin	186.21	0.42	55	Bryostatin	905
	8	0.64	57	Niclosamide (usan)	327.12	0.41	56	Glycoxalic acid	74.03
	9	0.63	46	Thiazolobenzimidazole	288.32	0.4	56	Fluorodopan	249.67
	10	0.63	56	Azd-4635	315.73	0.4	56	Merbarone	263.27

*p*—Pearson’s correlation coefficient (value ranges between −1 and 1 (values becomes more significant as they increase above)). CCLC—common cell lines count cell counts. MW—Molecular weight (g/mol).

**Table 4 cancers-13-02523-t004:** Common names, Uniprot and ChEMBL IDs, and target classes of specific gene targets of NSC765600.

Target	Common Name	Uniprot ID	ChEMBL ID	Target Class
Serine/threonine-protein kinase MTOR	MTOR	P42345	CHEMBL2842	Kinase
PI3-kinase p85-alpha subunit	PIK3R1	P27986	CHEMBL2506	Enzyme
Serotonin 2c (5-HT2c) receptor	HTR2C	P28335	CHEMBL225	Family A G protein-coupled receptor
Cyclin-dependent kinase 4	*CDK4*	P11802	CHEMBL331	Kinase
Matrix metalloproteinase 3	MMP3	P08254	CHEMBL283	Protease
Histone chaperone ASF1A	ASF1A	Q9Y294	CHEMBL3392950	Unclassified protein
PI3-kinase p110-delta subunit	PIK3CD	O00329	CHEMBL3130	Enzyme
Phosphodiesterase 5A	PDE5A	O76074	CHEMBL1827	Phosphodiesterase
CDK2/Cyclin A	CCNA2 CDK2	P20248 P24941	CHEMBL3038469	Kinase
Nuclear factor NF-kappa-B	NFKB1	P19838	CHEMBL3251	Other cytosolic protein
Cyclin-dependent kinase 2/cyclin E	CCNE2CDK2 CCNE1	O96020 P24941 P24864	CHEMBL2094126	Other cytosolic protein
Serine/threonine-protein kinase *PLK1*	*PLK1*	P53350	CHEMBL3024	Kinase
Beta-glucuronidase	GUSB	P08236	CHEMBL2728	Enzyme
cAMP-dependent protein kinase alpha-catalytic subunit	PRKACA	P17612	CHEMBL4101	Kinase
Interleukin-8 receptor B	CXCR2	P25025	CHEMBL2434	Family A G protein-coupled receptor
Dual specificity tyrosine-phosphorylation-regulated kinase 1B	DYRK1B	Q9Y463	CHEMBL5543	Kinase
Cyclin-dependent kinase 4/cyclin D1	*CCND1 CDK4*	P24385 P11802	CHEMBL1907601	Kinase
Cyclophilin A	PPIA	P62937	CHEMBL1949	Isomerase
Platelet-derived growth factor receptor alpha	PDGFRA	P16234	CHEMBL2007	Kinase
Glycogen synthase kinase-3 alpha	GSK3A	P49840	CHEMBL2850	Kinase

**Table 5 cancers-13-02523-t005:** Common names, Uniprot and ChEMBL IDs, and target classes of specific gene targets of NSC765691.

Target	Common Name	Uniprot ID	ChEMBL ID	Target Class
Serine/threonine-protein kinase	*PLK1*	P53350	CHEMBL3024	Kinase
Nicotinamide phosphoribosyl transferase	NAMPT	P43490	CHEMBL1744525	Enzyme
Rho-associated protein kinase 1	ROCK1	Q13464	CHEMBL3231	Kinase
Monoamine oxidase B	MAOB	P27338	CHEMBL2039	Oxidoreductase
Focal adhesion kinase 1	PTK2	Q05397	CHEMBL2695	Kinase
Vascular endothelial growth factor receptor 2	KDR	P35968	CHEMBL279	Kinase
Tyrosine-protein kinase TIE-2	TEK	Q02763	CHEMBL4128	Kinase
Cyclin-dependent kinase 5/CDK5 activator 1	CDK5R1 CDK5	Q15078 Q00535	CHEMBL1907600	Kinase
Cyclin-dependent kinase 7	CDK7	P50613	CHEMBL3055	Kinase
Platelet-derived growth factor receptor alpha	PDGFRA	P16234	CHEMBL2007	Kinase
TGF-beta receptor type I	TGFBR1	P36897	CHEMBL4439	Kinase
Phosphodiesterase 5A	PDE5A	O76074	CHEMBL1827	Phosphodiesterase
Cyclin-dependent kinase 4/cyclin D1	*CCND1 CDK4*	P24385 P11802	CHEMBL1907601	Kinase
Rho-associated protein kinase 2	ROCK2	O75116	CHEMBL2973	Kinase
Cyclin-dependent kinase 2	CDK2	P24941	CHEMBL301	Kinase
Cyclin-dependent kinase 1	CDK1	P06493	CHEMBL308	Kinase
Cyclin-dependent kinase 4	*CDK4*	P11802	CHEMBL331	Kinase
G-protein coupled receptor kinase 2	GRK2	P25098	CHEMBL4079	Kinase
Toll-like receptor (TLR7/TLR9)	TLR9	Q9NR96	CHEMBL5804	Toll-like and Il-1 receptors
Glycogen synthase kinase-3 beta	GSK3B	P49841	CHEMBL262	Kinase

## Data Availability

The datasets generated and/or analyzed in this study are available upon reasonable request.

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
