# Peer review of "Network Pharmacological Analysis through a Bioinformatics Approach of Novel NSC765600 and NSC765691 Compounds as Potential Inhibitors of CCND1/CDK4/PLK1/CD44 in Cancer Types"

_cancers, 2021, doi:10.3390/cancers13112523_

Round 1
Reviewer 1 Report
The study is innovative and interesting. It is still in an experimental phase and therefore needs further research to confirm its validity.
I think that the term "multiple cancer types" should be removed, since the pathogenetic mechanisms that regulate the progression of each single type of cancer are numerous and different from each other.
In this experimental study the authors, through a bioinformatics approach, evaluate the potential antitumor activity of two innovative pharmacological compounds, NSC765600 and NSC765691.
The data analysis, using computer simulation, demonstrated the potential pharmacological validity of these pharmacological compounds in terms of polarity, solubility, saturation, flexibility, and lipophilicity. Furthermore, both NSC765600 and NSC765691 exhibited BBB permeability.
Based on bioinformatics data, the authors identified as potential targets CCND1/CDK4/PLK1/CD44, as significantly expressed in various cancers and correlated with reduced survival.
The data obtained demonstrated an effective inhibitory activity against the identified targets. Specifically, antiproliferative and cytotoxic activity was highlighted in various tumor cell lines (melanomas, central nervous system cancers, renal cancer, breast cancer, NSCLC, leukemia, colon cancer, prostate cancer, and ovarian cancer).
The study appears interesting and certainly innovative. The scientific method adopted by the authors is consistent and the conclusions pertinent. The identified potential targets are valid.
However, it is only right to specify that this is a purely experimental study and that further studies will be necessary to confirm the validity of the data obtained.
Reviewer 2 Report
In order to develop new anticancer strategies, it is important to discover new anticancer drugs through a bioinformatics approach. This paper shows that two candidate small molecules, NSC765600 and NSC765691, were designed using comprehensive computational and bioinformatics analysis both of which met the drug-likeness criteria and had potential inhibitory effects on CCND/CDK4/PLK1/CD44 oncogenic pathway. Also, they showed satisfactory levels of safety with regards to toxicity.
Generally, the analyses were systematic and the paper reads well. However, the authors should carefully think about that all the results were based on simulation and computer-based calculation. The results obtained in the paper are merely theoretical ones even if the analyses were satisfactory ones. In the abstract there is a description “Currently, further in vitro and in vivo investigations in tumor-bearing mice are in progress to study the potential treatment efficacies of the novel NSC765600 and NSC765691 small molecules.”, but this does not guarantee the reliability of the development of these drugs, nor does it prove their effectiveness. Besides, actual IC50 values of those compounds against CCND/CDK4/PLK1/CD44 molecules should be measured in the experimental basis in the future. Actually there are still many walls to be overcome in the development of anticancer drugs. So, the authors should clearly mention about it in the text.
Reviewer 3 Report
The manuscript describes the general characterization of two small molecules derived from Diflunisal and Fostamatinib. However, some important points are incompletely and incorrectly described and there are a few issues that definitely need to address before publication.
- The authors selected 4 candidate genes from those predicted bycted by swisstargetprediction.ch. Most importantly, there needs to be information about the overall prediction by this tool. Were these the top 4 genes? What was the p-value/score for these genes compared to the distribution predicted for the other genes? What are the target predictions for the precursor drugs (Diflunisal and Fostamatinib)? Also, a couple of lines explaining how the tool arrives at these predictions or why the authors chose this tool would be helpful.
- The legend and data in Figure 3 don't seem to match. The legend states "Elevated mRNA levels of CCND1/CDK4/PLK1/CD44 oncogenic signaling were found to be associated with shorter survival percentages ..." but the hazard ratio for a and d is less than 1, while it is more than 1 for b and c. So elevated mRNA levels are associated with longer survival in CCND1 and CD44, and with shorter survival in PLK1 and CDK4. Also, individual subplot captions, for example a) states "The mRNA level of CCND1 increased at p=1.1e-08". This statement is both meaningless and incorrect. The p-value is the association of mRNA levels with survival, not a measure of increase. Also, exactly which cancer type from TCGA was chosen in the analysis needs to be written clearly.
- At least to me, it wasn't clear how Table 3 is calculated. What exactly are the fingerprints and mechanistic correlations? Are these based on predicted gene targets, physiochemical properties of Table 2, mutation and variant NCI cancer cell line growth rates, or something else?
- Again in Figure 10, the data isn't described completely and the conclusion/caption seems to be wrong. The authors seem to imply that the two small molecules show anticancer properties generally. However, it seems that Mean Growth Percent - Growth Percent is positive for some cancers and negative for some cancers. Seems like the correct conclusion is that the molecules show anti-cancer and pro-cancer properties in different cell lines.
- The authors should explain that the Mean Growth Percent in Fig. 10 refers to. Different growth percent values (column 2 of the figure) give different positive and negative scores (bar chart). How are these calculated. Fig. 10 (a) doesn't have mean, delta, and range values. How statistically significant are the positive and negative values? There needs to be a measure of how different the positive and negative values are compared to what is expected due to experimental noise.
Reviewer 4 Report
Reviewers report
Title: Network Pharmacological Analysis Through a Bioinformatics Approach of Novel NSC765600 and NSC765691 Compounds as Potential Inhibitors of CCND1/CDK4/PLK1/CD44 in Multiple Cancer types
Version: 1
Date: 13th April 2021
Reviewers report
The authors predict drug targets of NSC765600 and NSC765691 by comprehensive computational and bioinformatics analysis, as well as their own in silico molecular docking analysis. They showed ligand-protein interactions between the compounds with CCND1, CDK4, PLK1, and CD44, which are involved in tumor formation. Finally, they tried to demonstrate the effect of the compounds by using cell lines. The in silico strategy is intriguing, but some of results are not properly presented, and the effects of the compounds are not clearly shown in the final in vitro experiment. In addition, it is not clearly discussed on advantages of the compounds and comparison to the currently existed “standard” compounds. The authors need to revise the manuscript to show and discuss the results properly according to the following comments, and then the manuscript need to be carefully estimated again.
Major comments:
- Results 3.2, Table 4 and 5: No safety data of the compounds was shown in this manuscript. The authors claim NSC765600 and NSC765691 are potentially druggable genes. However, there is no information of the affinity and off-target effect of these drugs. Authors need to indicate values related to on/off target binding, hopefully which can be statistically evaluated, in Table 4 and 5.
- Materials and methods (MM) 2.6.; Results 3.6; Figure 10. The in vitro assay of anticancer activities using human tumor cell lines is a key of this study to assess effects of the compounds. The authors need to provide more detail of method: how estimate cytotoxic and anti-proliferative activity, why “mean percentage of treated tumor cell lines” but not the data from untreated cells were used for standard value. In Figure 10, it is not understandable the relationship between “growth percent” (left values) and “Mean Growth percent – Growth percent” (right bars), which are needed to be explained more. The authors also have to explain “delta” and “range”, and what can be interpreted from these values. In addition, the left values and right bars don’t match in Figure 10b, and it’s not clear the reason why -0.78 can be calculated as a percentage in HOP-92.
- Table 3 is really difficult to be understood, and has to be largely revised. Some columns are not aligned with values. No need to write g/mol with each value, since it is already mentioned in the head of the table. Probably two tables are combined, and then Target NSC and MW are needed for both right and left part of the compounds?
- Figure 3: Apparently, survival ratio of overexpression of CCND and CD44 overall look to be higher than that of low expression. Actually, HR (hazard ratio) were lower in overexpression of these genes. Then inhibition of the genes by the compounds cannot simply suppress tumor, right? The curves seem to be averages or some ways to be merged data of multiple cancer types, but if the survival curve would be changed among the types, probably they should not be merged and each curve should be separately shown? Then you might see the specificity to types of tumor? Nevertheless, the authors have to state these points clearly, and discuss them in term of effects of the compounds. In addition, there is no information about dot lines.
- Figure 4: Expression and IHC data for CDK4 are missing, although described in the title of Figure 4.
- Figure 6: the letters are too small, and the graph has to be revised to show all information clearly.
- Figures 7 and 8: The graphs are squeezed and it is very difficult to see the information. The graph has to be revised to show all information clearly.
- Discussion: the results doesn’t seem to be discussed properly. It looks like a shorter copy of Results section, and data interpretations are not really provided. For instance, the authors should discuss in which cancer types NSC765600 and NSC765691 would work better compared to the current drugs. In addition, the authors should discuss in a point of safety (off-target effects) of the compounds for clinical applications.
- No description/legend of Supplementary figures.
Minor comments:
- First part of Abstract is not cohesive: the link between CCND1, CDK4, PLK1, and CD44 and NSC765600 and NSC765691 is not emphasized.
- Introduction, line 4: complexities in cancer, including survival
- Introduction, line 5: resistance to current therapeutic interventions.
- Introduction, line 9-10: Accumulating evidence showed, that another mechanism through which cancer develops and progresses results from changes in the cell cycle.
- Introduction, line 13: has been reported
- Introduction, line 22: Therefore, novel therapeutic approaches exhibiting better responses are needed.
- Introduction, line 26: has been reported
- MM 2.5, line 1: The in bold
- MM 2.5, line 3: PLK1, respectively
- Figure 1: NSC765600 (a) and NSC765691 (b).
- Results 3.2, line 1: prediction
- Results 3.3, line 3: tumor tissues, compared to
- Results 3.3, line 12: immunohistochemistry
- Figure 2 and its description were separated with some other text. Figure 2a, b, c, d should be separated.
- Discussion, line 28: online prediction tools
- If your table is divided to 2 pages, it is always good to make the head for both parts (Table 4,5).
Round 2
Reviewer 4 Report
The manuscript has been greatly improved.